# Guava Seed Oil: Potential Waste for the Rhamnolipids Production

Ingrid Yoshimura [1], Ana Maria Salazar-Bryam [1], Adriano Uemura de Faria [1], Lucas Prado Leite [1], Roberta Barros Lovaglio [2] and Jonas Contiero [1,3,*]

[1] Institute of Biosciences, São Paulo State University (Unesp), Rio Claro 13506-900, SP, Brazil
[2] Center of Natural Sciences, Federal University of São Carlos (UFSCar), Lagoa do Sino Campus, Buri 18290-000, SP, Brazil
[3] Institute for Research in Bioenergy, São Paulo State University (Unesp), Rio Claro 13500-230, SP, Brazil
* Correspondence: jonas.contiero@unesp.br; Tel.: +55-19-3526-4149

**Abstract:** Guava is consumed in natura and is also of considerable importance to the food industry. The seeds and peel of this fruit are discarded, however, guava seeds yield oil (~13%) that can be used for the bioproducts synthesis. The use of a by-product as a carbon source is advantageous, as it reduces the environmental impact of possible harmful materials to nature, while adding value to products. In addition, the use of untested substrates can bring new yield and characterization results. Thus, this research sought to study rhamnolipids (RLs) production from guava seed oil, a by-product of the fructorefinery. The experiments were carried out using *Pseudomonas aeruginosa* LBI 2A1 and experimental design was used to optimize the variables Carbon and Nitrogen concentration. Characterization of RLs produced occurred by LC-MS. In this study, variables in the quadratic forms and the interaction between them influenced the response ($p < 0.05$). The most significant variable was N concentration. Maximum RLs yield achieved 39.97 g/L, predominantly of mono-RL. Characterization analysis revealed 9 homologues including the presence of $RhaC_{10}C_{14:2}$ (*m/z* 555) whose structure has not previously been observed. This research showed that guava seed oil is an alternative potential carbon source for rhamnolipid production with rare rhamnolipid homologues.

**Keywords:** biosurfactants; C/N ratio; congeners characterization; experimental design; glycolipids; mass spectrometry; *Pseudomonas aeruginosa*

## 1. Introduction

Biosurfactants are compounds with considerable potential to replace chemical surfactants derived from the oil industry [1]. These natural compounds are synthesized by microorganisms, and have advantages, such as high biodegradability, low toxicity, production from renewable sources [2], a higher degree of specificity, unique structures, high selectivity and stability in situations of extreme temperature and high salinity [3–11]. The amphiphilic chemical-structural conformation results in reduce surface and interfacial tension and have detergency, solubilization, emulsification, lubrication and phase dispersion properties [12,13]. Biosurfactants are generally are more efficient, with a critical micelle concentration (CMC) lower (range: 5 to 380 mg/L) than that of chemical surfactants [3], which justifies the development studies on this product.

Their versatility constitutes another relevant factor, with numerous industrial applications due its surface activity and antimicrobial properties [14,15]. As a practical example, biosurfactants can assist in the bioremediation, fragmentation and dispersion process in situations involving oil spills and contamination by metals [13,16,17]. Their use as nanoparticle stabilizers, such as nZVI, has been extensively studied, mainly for the removal of contaminants, such as heavy metals, nitrate and nitrite, from groundwater due to the reduction in the oxidation of these pollutants [18–26]. They also can also be used in the pharmaceutical, cosmetic, oil and agricultural industries for solubilization and emulsification, among other processes [14,27–29].

Rhamnolipids (RLs) constitute a biosurfactant category belonging to the group of glycolipids. These compounds are produced mainly by strains of the bacterium *Pseudomonas aeruginosa*. The structure is formed by one or two rhamnoses and fatty acids [12,30]. Rhamnolipids are composed of a mixture of homologues, in which mono-rhamnolipids (main congener $RhaC_{10}C_{10}$) and di-rhamnolipids (main congener $Rha_2C_{10}C_{10}$) are predominant [31,32]. The advantages of rhamnolipids are the high surface activity and high production yield [33]. However, the production of biosurfactants is expensive, which makes it difficult for these natural compounds to compete with chemical surfactants and gain ground in the market. For application in industries, bioproducts must be produced profitably and on a large scale.

According to Mukherjee et al. [34], the economy of the production of biotechnological products is governed by variables such as the enhancement of the bioprocess, the purpose of which is to find the ideal conditions for the cultivation of the microorganism to obtain the highest yield of the biotechnological product. Factors such as the concentration of nitrogen and iron, agitation, temperature and pH directly affect microbial growth and the production of metabolites [3,34,35]. Traditionally, the selection of these parameters is achieved by varying each factor individually in order to select optimal points. However, statistical optimization can be achieved using the response surface methodology to determine interactions between factors and reduce the number of experiments [36]. Another important aspect is the cost of raw materials, as the substrate accounts for up to 30% of the total production of a biosurfactant [34,37]. Thus, cheaper equipment and substrates are needed. The use of a waste product or byproduct as a carbon source is advantageous, as it reduces the environmental impact of materials that can be considered harmful to nature while adding value to products that would otherwise be discarded. In addition to cost reduction, the use of byproducts as carbon sources may also increase production yields. Fruit bagasse of yellow cashew [38], beet husk [39], frying oil [4], residual biodiesel glycerol [40], sugar beet, banana, citrus, fried palm oil, moringa and yam residues [41] and mango seed oil [42] are examples of carbon sources used for this purpose.

Thus, various of agro-industrial byproducts are promising candidates that can assist in lowering the cost of biosurfactant production. Guava is a widely appreciated fruit. In Brazil, the commercial prospects of this fruit are enhanced by the favorable climate and soils, offering profitability and development to agricultural production [43]. In 2018, Brazil produced 1,897,904 tons of guava, mango and mangosteen, ranking fifth in the world in this category [44]. In addition to being consumed in natura, guava is also of considerable importance to the food industry, where it plays the role as raw material in the production of juices, jellies, syrups, sweets, etc. However, this fruit processing scenario also results in the disposal of material considered waste, such as fruit peels and seeds. The portion that is not used is called a co-product and leads to problems related to disposal and environmental pollution [45].

Guava seeds correspond 6 to 12% of the total weight of the fruit and account for the largest portion of waste. These seeds are discarded in landfills, contributing to environmental problems. However, this byproduct of fruit processing can offer a financial advantage, as it contains oil that may have diverse applications. Guava seeds yield 10 to 16% oil, depending on the growth and processing conditions [46–48]. This oil could be used for production of rhamnolipids. Studies have been conducted on the production of rhamnolipids using hydrophilic and hydrophobic carbon sources. Though, hydrophobic sources appear to be the better substrate [7]. In addition, guava seed oil is composed of polyunsaturated fatty acids (PUFA)—predominantly linoleic acid (omega-6)—and monounsaturated fatty acids (MUFA)—predominantly oleic acid (omega-9). The oil has low acidity and peroxide values. Its antioxidant activity is close to 59%. It also has sufficient amounts of tocopherol and carotenoids and more than 30 volatile constituents, mainly esters of fatty acids [48]. This alternative carbon source from the fruit refinery process has not previously been used to produce a biosurfactant. Therefore, the aim of this work was to evaluate the use of guava seed oil as a substrate for the production of rhamnolipids by *Pseudomonas aeruginosa*

LBI 2A1. Optimization of the cultivation conditions was performed using the response surface methodology. The rhamnolipids produced were extracted and characterized by mass spectrometry.

## 2. Materials and Methods

### 2.1. Rhamnolipid Production

#### 2.1.1. Microorganism

The strain *Pseudomonas aeruginosa* LBI 2A1 [49] was used in these studies for the production of rhamnolipids. The culture media and growth conditions were those previously described by Salazar-Bryam et al. [40]. The microorganism came from a stock culture in a cryogenic tube composed of lysogenic broth with 20% glycerol and stored in an ultrafreezer at $-80\ ^\circ$C.

#### 2.1.2. Culture Medium

The composition of the medium used for the pre-culture of the microorganism was 10 g/L of tryptone, 5.0 g/L of yeast extract and 10 g/L of NaCl [50]. The mineral medium was used for the inoculum and fermentation production in Erlenmeyer flasks was composed of (g/L) $MgSO_4$ $7H_2O$ (0.5), KCl (1), $K_2HPO_4$ (0.3), $NaNO_3$ and guava seed oil (both concentrations varied with the experimental design) as well as 1 mL/L of the trace element solution. The trace element solution was composed of (g/L) $Na_3C_6H_5O_7$ $2H_2O$ (2.0), $FeCl_3$ $6H_2O$ (0.28), $ZnSO_4$ $7H_2O$ (1.4), $CoCl_2$ $6H_2O$ (1.2), $CuSO_4$ $5H_2O$ (1.2) and $MnSO_4$ $H_2O$ (0.8). The initial pH of the medium was adjusted to 6.8–7 using a 1M NaOH and HCl solution [50,51].

#### 2.1.3. Carbon Source—Guava Seed Oil

Guava seeds (20 kg) were kindly provided by Indústria Predilecta Alimentos located in São Lorenço do Turvo, Matão, State of São Paulo, Brazil. Oil extraction was carried out at the AGTTEC Coffee Processing and Equipment Maintenance Laboratory LTD using a multi-product cold extraction press (Agitec brand, model PF-AG2), with a capacity of 20 kg/hour powered by a 2CV motor.

#### 2.1.4. Cultivation Conditions

The microorganism was pre-cultured in Erlenmeyer flasks (125 mL) containing 20 mL of lysogenic broth and 100 μL of *P. aeruginosa* LBI 2A1 stock culture. Incubation was performed in a Multitron Standard Infors HT shaker for 24 h at 37 $^\circ$C and 200 rpm. Next, 2.5 mL of the fermented broth were transferred to the inoculum containing mineral medium and half the concentration (50 g/L) of the carbon source used in the fermentation medium. Cultivation took place in 500-mL Erlenmeyer flasks containing 100 mL of medium under the same conditions mentioned above. Absorbance readings ($\lambda = 580$ nm) were taken using a Bel Spectro SP-2000 spectrophotometer. These measurements were used to inoculate the fermentation media, resulting in an initial optical density of 0.1. Fermentations were also carried out in 500-mL Erlenmeyer flasks containing 100 mL of mineral medium. In these cultures, the production of rhamnolipids and consumption of the carbon source were analyzed by taking samples every 24 h. The experiments were carried out in triplicate in a shaker at 200 rpm and 37 $^\circ$C for 96 h. The trace element solution was added at 0, 20, 40 and 70 h based on Müller et al. [50,51].

### 2.2. Optimization of Rhamnolipid Production

An experimental design was used to select the best conditions for the maximization of rhamnolipid production with the carbon source that demonstrated the greatest viability. A first experimental step provided the basis for establishing the best range to be used in the central composite rotational design (CCRD) and response surface methodology. The independent variables were carbon source concentration (guava seed oil; $X_1$) and nitrogen source concentration (sodium nitrate; $X_2$). CCRD $2^2$ was performed with four axial points

and four repetitions at the central point, totaling 12 trials. Table 1 displays the design with the conditions used and presents the values studied.

**Table 1.** CCRD $2^2$ planning matrix with growing conditions in coded and real levels of carbon source ($X_1$) and nitrogen source ($X_2$).

| Essay | Carbon Source ($X_1$) | | Nitrogen Source ($X_2$) | |
|---|---|---|---|---|
| | Coded Level | Real Values (g/L) | Coded Level | Real Values (g/L) |
| E1 | −1 | 50 | −1 | 5 |
| E2 | −1 | 50 | 1 | 15 |
| E3 | 1 | 150 | −1 | 5 |
| E4 | 1 | 150 | 1 | 15 |
| E5 | −1.414 | 29.3 | 0 | 10 |
| E6 | 1.414 | 170.7 | 0 | 10 |
| E7 | 0 | 100 | −1.414 | 2.93 |
| E8 | 0 | 100 | 1.414 | 17.07 |
| E9 | 0 | 100 | 0 | 10 |
| E10 | 0 | 100 | 0 | 10 |
| E11 | 0 | 100 | 0 | 10 |
| E12 | 0 | 100 | 0 | 10 |

The "RL production" response was modeled using the second-order polynomial Equation (1):

$$y = \beta_0 + \beta_{1\times1} + \beta_{11}X_{12} + \beta_2X_2 + \beta_{22}X_{22} + \beta_{12}X_1X_2 \ldots \beta_kX_k + \beta_{kk}X_{k2}. \quad (1)$$

in which $y$ is the predicted response, $\beta_0$ is the intercept term, $\beta_1$ and $\beta_2$ are the linear effects, $\beta_{11}$ and $\beta_{22}$ are the quadratic effects, $\beta_{12}$ is the interaction effect, $X_1$ and $X_2$ are the independent variables and k is the number of independent variables.

### 2.3. Sample Processing

To analyze the biosurfactant production and quantify the biomass, 10-mL samples were taken, to which the same volume of n-hexane was slowly added. The samples were processed in a Hitachi CR22G centrifuge at 10,000× *g* and 10 °C for 10 min for separation into three phases: cells/aqueous phase/organic phase. The hydrophilic portion and the pellet were used to quantify rhamnolipids and biomass, respectively. The organic phase containing n-hexane was used to quantify the residual oil.

### 2.3.1. Determination of Microbial Biomass

The evaluation of microbial growth was performed using the gravimetric method. The cell pellet was resuspended in 0.85% (*v*/*v*) NaCl (same volume as that of the initial sample) and centrifuged at 10,000× *g* and 10 °C for 10 min. The resulting pellet was dissolved in a smaller volume of distilled water and placed into a FANEM drying oven (mod. 315/4) at 100 °C until reaching a constant weight [50,51].

### 2.3.2. Determination of Consumption of Carbon Source

Residual oil was quantified after the removal of the organic phase for gravimetric determination. Samples were kept at room temperature inside a fume hood until the complete evaporation of the solvent and then weighed.

Aliquots were submitted to a transesterification reaction (Official AOAC Method 969.33, 1969), starting from 100 mg of residual oil and 4 mL of 0.5 M NaOH in MeOH. The solution was refluxed and maintained for approximately 10 min. Next, 5 mL of 10% BF$_3$ in MeOH were added through the condenser and the system was boiled for another 2 min. Lastly, heptane (2 mL) was added also through the condenser, with subsequent boiling for 1 min. The system was removed from reflux with the interruption of the heat and removal of the condenser. Next, 15 mL of saturated NaCl solution were added and the mixture

was stirred vigorously for 15 s. The solution was transferred to a test tube for gravimetric separation of the organic phase, which was removed, dried with $Na_2SO_4$.

The analysis of the composition of the residual oil was performed in a GC-MS coupled to a Shimadzu GCMS-QP 2010 Ultra electron impact mass spectrometer. The RTX-5MS fused silica column (30 m, 0.25 mm, 0.25 μm) was used. Oven temperature was 120 °C, maintained for 1 min, with an increment of 20 °C/min to 170 °C, 3 °C/min to 210 °C and 20 °C/min to 250 °C, then held constant for 10 min. The injector temperature was 250 °C. Injection was performed in split mode (1:10 and 1:100) with an injection volume of 1 μL and the carrier gas was helium with a flow of 1 mL/min. The range of masses analyzed by the mass spectrometer was 41 to 350, starting at 2.5 min until the end of the run (28.8 min). For the identification of fatty acids, the retention times of the peaks of the sample chromatograms were compared to pure standards of fatty acid methyl esters (37-Component FAME Mix, Sulpelco®).

### 2.4. Extraction of Rhamnolipids

An 85% $H_3PO_4$ 1:100 (*v/v*) and ethyl acetate 1:1.25 (*v/v*) solution was added to the hydrophilic portion, which enabled the precipitation and extraction of the surfactant. Samples were centrifuged at 10,000× *g* and 10 °C for 10 min. Extraction with ethyl acetate was performed twice [50,51].

### 2.4.1. Determination of Rhamnolipid Concentration—HPLC

The patterns and aliquots of rhamnolipids were analyzed by thin layer chromatography [52] and derivatized as described by Schenk et al. [53] to be determined with a via UV detector. For such, samples containing rhamnolipids in ethyl acetate were evaporated. Next, 360 μL of acetonitrile and 40 μL of a mixture composed of 135 mM 4-bromophenacyl bromide/67.5 mM triethylamine (1:1) (*v/v*) were added. Derivatization was carried out for 1.5 h at 60 °C and 1200 rpm in a dry bath with AG-100 Agimaxx agitation.

Quantitation was performed with a Shimadzu high-performance liquid chromatography (HPLC) device coupled to a UV SPD 20-A detector, as described by [50]. For HPLC calibration, a standard solution of mono- and di-rhamnolipid was used at concentrations of 1.0, 0.5, 0.25, 0.125 and 0.0625 g/L. The reversed phase column NST 18 100A—C18 (5 μm, 150 × 4.6 mm) was used at 30 °C for the stationary phase. The mobile phases were Solution A—ultrapure water/methanol (95:5) (*v/v*) and Solution B—methanol/ultrapure water (95:5) (*v/v*). The flow was 0.4 mL/min for 35 min. Retention times were 22.4 ± 0.1 min for $RhaRhaC_{10}C_{10}$ and 23.3 ± 0.1 min for $RhaC_{10}C_{10}$.

### 2.4.2. Determination of Composition of Homologues by LC-MS

Rhamnolipids samples were resuspended in acetonitrile, diluted in ultrapure water/acetonitrile (7:3) (*v/v*) to a final concentration of 500 ppm and filtered through a 0.22-μm membrane. Analyses were performed using a liquid chromatograph-mass spectrometer (LC-MS) coupled to a Waters Xevo TQD Triple Quadrupole electrospray ionization mass spectrometer (ESI-MS) in negative mode. The apparatus was equipped with the Acquity UPLC Beh-C18 reversed-phase column (1.7 μm, 2.1 x 50.0 mm; Waters Corp.).

In the LC analysis, mobile phases composed of 0.1% formic acid-water (A) and 0.1% formic acid-acetonitrile (B) were used. The gradient was 0/20, 20/95, 35/95, 36/20 (min/%B), the flow rate was 0.4 mL/min, the injection volume was 10 μL and run time was 32 min [54]. The mass range analyzed was from 200 to 800 *m/z*. For the MS/MS experiments, the gas was argon and the collision energy ranged from 5 to 25 eV.

## 3. Results and Discussion

The study carried out using the central composite rotational design with the independent variables (concentration of nitrogen source [sodium nitrate] and carbon source [guava seed oil]) and RL production are shown in Table 2. The results refer to the production of RL for 96 h of fermentation time, when production was maximum for most of the trials.

During this time, biosurfactant production ranged from 11.04 to 42.15 g/L (assays E1 and E11, respectively). Thus, the variation in the concentration of carbon and nitrogen exerted an influence on the production of the metabolite.

**Table 2.** Central composite rotational design used in cultivation of *P. aeruginosa* LBI 2A1. Media were composed of guava seed oil as carbon source and sodium nitrate as nitrogen source. Independent variables represented by coded and real levels. Responses correspond to 96 h of fermentation and show rhamnolipids, biomass, product/biomass conversion factor and carbon/nitrogen ratio in each assay.

| Essays | Coded Levels | | Real Levels (g/L) | | RL ($Y$) (g/L) | Biomass (g/L) | Conversion Factor Product/Biomass (g/g) | Proportion C/N |
|---|---|---|---|---|---|---|---|---|
| | C ($X_1$) | N ($X_2$) | C ($X_1$) | N ($X_2$) | | | | |
| E1 | −1 | −1 | 50.00 | 5.00 | 11.05 | 2.09 | 5.29 | 10.00 |
| E2 | −1 | 1 | 50.00 | 15.00 | 26.66 | 4.83 | 5.52 | 3.30 |
| E3 | 1 | −1 | 150.00 | 5.00 | 19.55 | 5.15 | 3.80 | 30.00 |
| E4 | 1 | 1 | 150.00 | 15.00 | 12.97 | 3.39 | 3.83 | 10.00 |
| E5 | −1.41 | 0 | 29.30 | 10.00 | 24.33 | 4.54 | 5.36 | 3.00 |
| E6 | +1.41 | 0 | 170.70 | 10.00 | 33.73 | 4.75 | 7.10 | 17.10 |
| E7 | 0 | −1.41 | 100.00 | 2.93 | 13.33 | 5.06 | 2.63 | 33.90 |
| E8 | 0 | +1.41 | 100.00 | 17.70 | 12.98 | 4.80 | 2.70 | 5.90 |
| E9 | 0 | 0 | 100.00 | 10.00 | 40.71 | 5.32 | 7.65 | 10.00 |
| E10 | 0 | 0 | 100.00 | 10.00 | 37.64 | 4.40 | 8.55 | 10.00 |
| E11 | 0 | 0 | 100.00 | 10.00 | 42.15 | 4.64 | 9.08 | 10.00 |
| E12 | 0 | 0 | 100.00 | 10.00 | 39.38 | 5.16 | 7.63 | 10.00 |

Trials E9–E12 had the best yield (39.97 g/L of RL, with C/N ratio of 10). However, Trials E1 and E4 resulted in the same C/N ratio of 10, but had the lowest RL yields (11.05 and 12.98 g/L, respectively). These essays had carbon and nitrogen concentrations at coded levels −1 or +1 simultaneously, which demonstrates that these variables at very low or very high levels—even in equilibrium—result in lower product formation. Therefore, the interaction between the two variables was not the most important factor to optimizing the response and yield.

The effects that each variable and its interactions exerted on RL production are shown in the Pareto diagram (Figure 1). Parameters with *p*-values < 0.05 were considered significant. Thus, the quadratic effects of the nitrogen and carbon sources as well as the interaction between the variables were significant. Nitrogen concentration was the factor with the greatest influence on RL production, followed by carbon concentration and, lastly, the interaction between the two variables.

According to Raza et al. [55] and Wu et al. [56], high proportions of C/N are beneficial to the production of RL. In studies by Guerra-Santos et al. [57], the limitation of nitrate resulted in a change in microbial metabolism, a reduction in the concentration of biomass and an increase in the production of the metabolite. Likewise, RL production increased as nitrogen was depleted in the study by for Ramana and Karanth [58]. In contrast, maximum productivity of RL occurred concomitantly with the excess of nitrogen in experiments carried out in a reactor by Lovaglio [6]. This suggests that the C/N ratio is not a standard to be followed and should not be taken as a singular or priority factor to optimize the production of rhamnolipids. The circumstances demonstrated in this experimental design suggest that keeping the nitrogen concentration within an optimized range is more important than the C/N ratio itself.

Analysis of variance (ANOVA) confirmed the data from the Pareto Diagram. Moreover, the correlation coefficient ($R^2$) for the production of RL was 0.944, which indicates that the experiments fit 94.4% to the proposed model (Table 3).

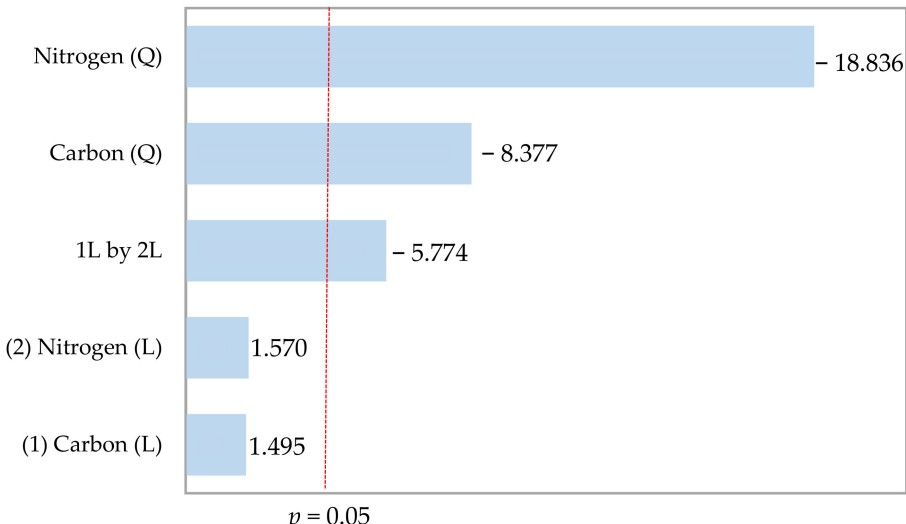

$$p = 0.05$$

**Figure 1.** Pareto diagram and estimated effects of carbon and nitrogen sources on production of RL by *P. aeruginosa* LBI 2A1 in 96 h of fermentation. Linear terms associated with letter L and quadratic terms with letter Q. Values considered absolute. 1L by 2L indicates interaction between factors.

**Table 3.** Analysis of variance (ANOVA) for production of RL by *P. aeruginosa* LBI 2A1 using central composite rotational design.

| Variable | Quadratic Sum | ANOVA Degrees of Freedom | Mean Square | F | *p* |
|---|---|---|---|---|---|
| (1) Carbon (L) | 8.239 | 1 | 8.239 | 2.2351 | 0.231785 |
| Carbon (Q) | 258.699 | 1 | 258.699 | 70.1796 | 0.003567 |
| (2) Nitrogen (L) | 9.095 | 1 | 9.095 | 2.4673 | 0.214267 |
| Nitrogen (Q) | 1307.878 | 1 | 1307.878 | 354.7991 | 0.000327 |
| 1L by 2L | 122.933 | 1 | 122.933 | 33.3490 | 0.010324 |
| Lack of Adjustment | 78.904 | 3 | 26.301 | 7.1350 | 0.070406 |
| pure error | 11.059 | 3 | 3.686 | | |
| Total Quadratic Sum | 1619.716 | 11 | | | |

Linear terms associated with letter L and quadratic terms with letter Q. 1L by 2L indicates interaction between factors.

From the RL production response (*y*), it was possible to model Equation (2):

$$y = 39.97 - 26.36\, X_1{}^2 - 14.29\, X_2{}^2 - 5.54\, X_{1\times2} \qquad (2)$$

as a function of the coded independent variables: $X_1$—carbon source (g/L); $X_2$—nitrogen source (g/L).

The response surface (Figure 2A) and contour curve (Figure 2B) graphs generated by the model demonstrate the interaction effects of the variables and ideal levels of each. The purpose was to identify the best cultivation conditions for maximum production of RL by *P. aeruginosa* LBI 2A1. The optimized concentration range was 75 to 135 g/L for the carbon source and 8.0 to 12.0 g/L for the nitrogen source.

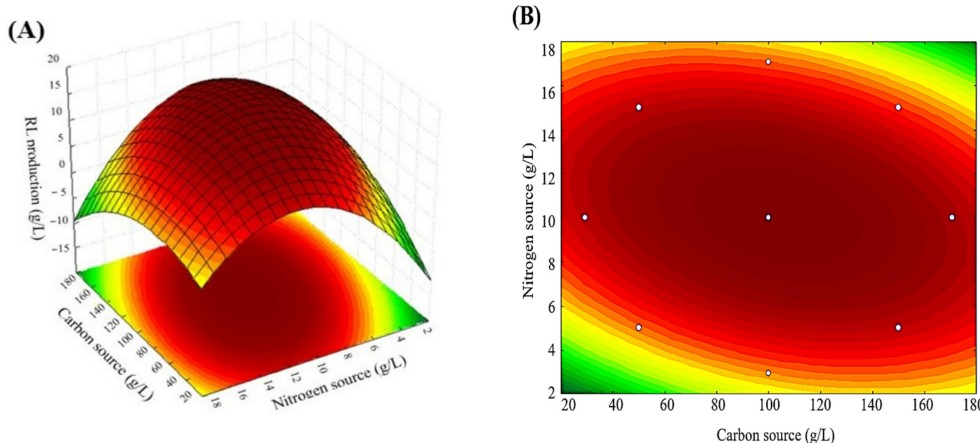

**Figure 2.** Response surface (**A**) and contour curve (**B**) for RL production by *P. aeruginosa* LBI 2A1 as function of carbon and nitrogen sources.

The graphs show that the excessive increase in these variables did not result in an increase in the production of the metabolite. The concentrations chosen for the central point (coded variables = 0) constituted the optimized region. Thus, trials 9 to 12 achieved the best results in the evaluation of rhamnolipid production by *P. aeruginosa* LBI 2A1 from the optimized conditions generated using the response surface methodology. And given that, these trials were better analyzed. Microbial growth, carbon source consumption, the production of mono- ($RhaC_{10}C_{10}$) and di- ($RhaRhaC_{10}C_{10}$) rhamnolipid homologues and total RLs are shown in Figure 3.

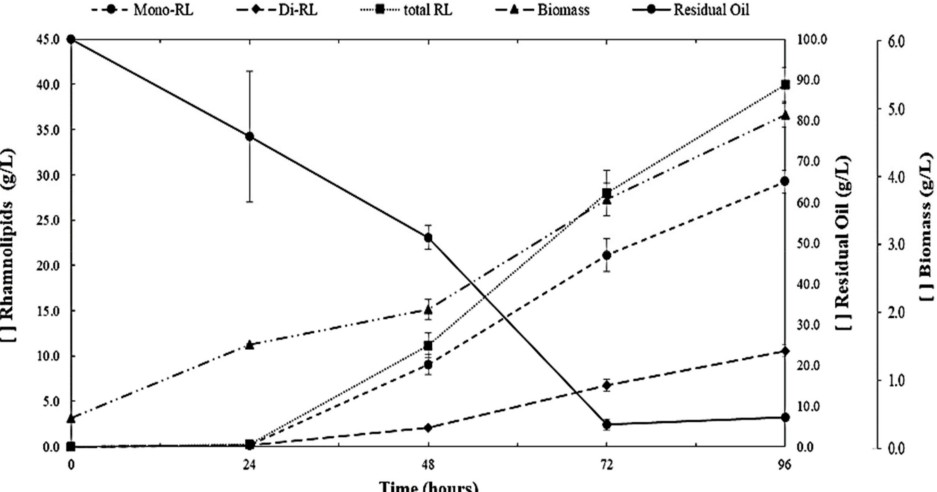

**Figure 3.** Growth, production of metabolites and carbon source consumption by *P. aeruginosa* LBI 2A1 in Trials 9–12 (coded variables = 0) of rotational central composite design. Cultivation took place at 200 rpm and 37 °C for 96 h of fermentation.

The cultivation conditions enabled microbial growth throughout the fermentation process, with a maximum biomass of $4.88 \pm 0.43$ g/L in 96 h. The amounts of the carbon and nitrogen sources were sufficient to maintain the bacterium in the exponential phase. The production of rhamnolipids was shown in this case to be partially associated with growth, as the rhamnolipid production began 24 h after the growth of the biomass.

This finding is in disagreement with data described by Radzuan et al. [59], who observed *P. aeruginosa* PAO1 growing in palm fatty acid distillate and fatty acid methyl ester as carbon sources. Rhamnolipids are typically produced in the stationary phase of fermentation, when the carbon source was still available. Sodagari1 and Ju [60] observed rhamnolipid production with different nitrogen concentrations and found that rhamnolipid

production was partially associated with growth using the highest concentration of nitrogen and associated with growth using the lowest concentration.

The concentration of rhamnolipids at 48 h (11.12 $\pm$ 1.37 g/L) likely assisted the bacterium to access the carbon source and enabled better growth. This may explain the greater slope of the biomass line from 48 to 72 h (slope = 0.51 at 24 h, 1.63 at 48 h and 1.24 at 72 h) as well as the sharpest decrease in oil consumption in the same period. According to Francy et al. [61] and Kappeli and Fiechter [62], one of the biological reasons for microorganisms to produce biosurfactants is related to the access to non-water-soluble substrates through the solubilization and emulsification of hydrocarbons. The production of the surfactant likely facilitated the assimilation, transport and metabolism of the oil by the bacterium, which justifies the correlation between RL synthesis and microbial growth. Moreover the product/biomass conversion factor increased throughout the experiment [data not shown], with a maximum value of 8.19 g/g at 96 h. These results are satisfactory and suggest that an increase in fermentation time may lead to even higher yields.

Another issue is the proportion of homologues species. The percentage of mono-rhamnolipid in 24h was 42.84 followed by 80.98 > 75.44 > 73.14 through the time. Between 48 and 96 h, a decrease in this percentage and an increase in the percentage of di-rhamnolipid occurred. This suggests that the second rhamnosyltransferase encoded by rhlC, which is regulated by rhlAB after the first 24 h [63], begins its action again, but without affecting the production of mono-rhamnolipid, showing a preference for the production of this congener in the case of the carbon source employed in this study.

The production of mono-RL was much higher. According to İkizler et al. [64] and Wu et al. [65], the fact that mono-RL has only one rhamnose modifies the arrangement of the layer available at the oil/water interface and results in a higher surface activity compared to di-RL. Such properties likely make mono-RL more beneficial to the microorganism. This direction of metabolism may be related to the fact that most of fatty acids in guava seed oil are unsaturated and contain long chain fatty acids. Nicolò et al. [66] demonstrated the influence of carbon sources in the homologues proportion. In their research, hydrophobic carbon source with high amount of long chain fatty acids—*Brassica* oil—resulted in higher proportion of mono-rhamnolipids. On the other hand, hydrophilic carbon sources—glucose and glycerol—and hydrophobic with short chain fatty acid showed a high production of di-rhamnolipids. The authors also investigated the transcriptional expression of *rhlC* gene, which was delayed in the first case compared to glycerol.

The characterization of guava seed oil at initial time showed 10 fatty acids. The major compounds were the long chain fatty acids: Linoleic (C18:2) 80.53% > Stearic (C18:0) 11.73% > Palmitic (C16:0) 6.26% > Arachidic (C20:0) 1.49%. From that, the fatty acid consumption during rhamnolipid production was also analyzed and showed in Figure 4. The fatty acid C20:0 was consumed quickly (within 24 h), followed by the consumption of C18:0 and C16:0, with similar kinetics (70% in the first 24 h). C18:2 was another fatty acid with high consumption in the first 24 h (50%), showing the preference in terms of fatty acids by the microorganism: C20:0 > C18:0, C16:0 > C18:2.

The LC-MS analyses revealed the composition of homologues species produced by *P. aeruginosa* LBI 2A1 using guava seed oil as substrate (Table 4). The ions found showed *m/z* 475.6 (RhaC$_8$C$_{10}$/RhaC$_{10}$C$_8$), 503.4 (RhaC$_{10}$C$_{10}$), 529.2 (RhaC$_{12:1}$C$_{10}$/RhaC$_{10}$C$_{12:1}$), 530.9 (RhaC$_{12}$C$_{10}$), 555.6 (RhaC$_{10}$C$_{14:2}$), 621.4 (RhaRhaC$_{10}$C$_8$/RhaRhaC$_8$C$_{10}$), 649.1 (RhaRhaC$_{10}$C$_{10}$), 675.6 (RhaRhaC$_{10}$C$_{12:1}$/RhaRhaC$_{12:1}$C$_{10}$) and 677.6 (RhaRhaC$_{10}$C$_{12}$/RhaRhaC$_{12}$C$_{10}$). These ions were fragmented to confirm that they were homologous species of rhamnolipids (Figure S1).

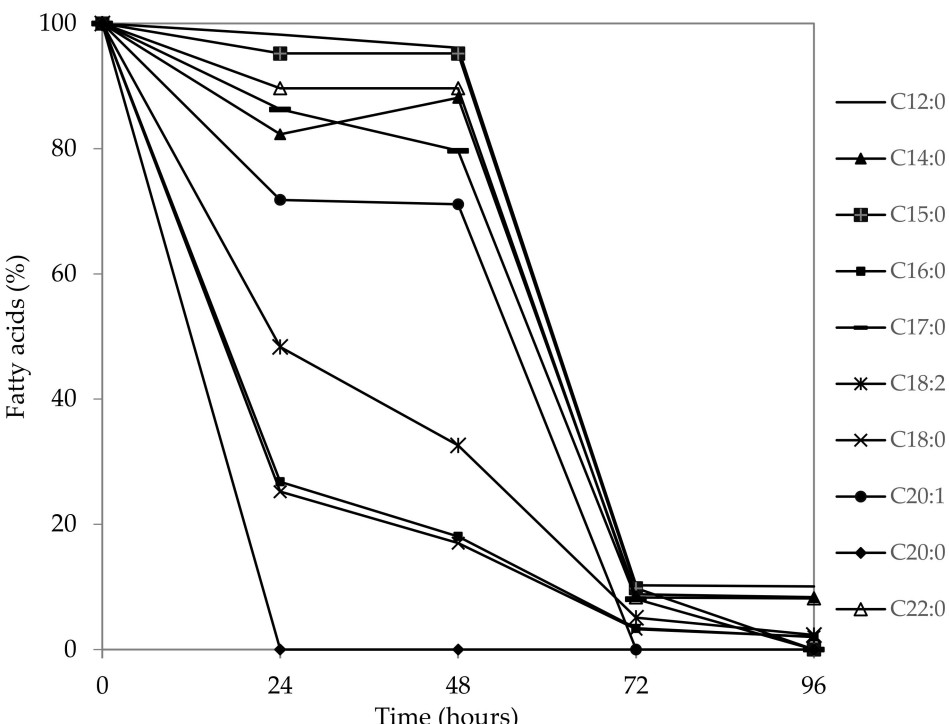

**Figure 4.** Consumption of fatty acids from residual guava seed oil by *P. aeruginosa* LBI 2A1. Cultivation took place at 200 rpm and 37 °C for 96 h of fermentation.

**Table 4.** Homologues found in rhamnolipid produced by *P. aeruginosa* LBI 2A1 using guava seed oil as substrate.

| No. | Elementary Composition | Molecular Structure | [M-H]⁻ m/z | Retention Time [min] | Relative Abundance (%) |
|---|---|---|---|---|---|
| I | $C_{24}H_{44}O_9$ | RhaC$_8$C$_{10}$ | 475.6 | 10.12 | 11.84 |
| II | $C_{26}H_{48}O_9$ | RhaC$_{10}$C$_{10}$ | 503.4 | 12.09 | 100.00 |
| III | $C_{28}H_{50}O_9$ | RhaC$_{10}$C$_{12:1}$ | 529.2 | 13.27 | 43.44 |
| IV | $C_{28}H_{52}O_9$ | RhaC$_{10}$C$_{12}$ | 530.9 | 14.29 | 48.63 |
| V | $C_{30}H_{51}O_9$ | RhaC$_{10}$C$_{14:2}$ | 555.6 | 14.13 | 9.30 |
| VI | $C_{30}H_{54}O_{13}$ | RhaRhaC$_{10}$C$_8$; RhaRhaC$_8$C$_{10}$ | 621.4 | 9.22; 9.33 | 7.59 |
| VII | $C_{32}H_{58}O_{13}$ | RhaRhaC$_{10}$C$_{10}$ | 649.1 | 11.12 | 28.37 |
| VIII | $C_{34}H_{60}O_{13}$ | RhaRhaC$_{10}$C$_{12:1}$ | 675.6 | 12.28 | 9.41 |
| IX | $C_{34}H_{62}O_{13}$ | RhaRhaC$_{12}$C$_{10}$; RhaRhaC$_{10}$C$_{12}$ | 677.6 | 13.13 | 57.24 |

Designation Cx:n means fatty acid chain with chain length of x and n unsaturated bonds (–2n H). Retention time refers to times found in scan performed in selected ion recording mode.

Using soybean oil sludge as a carbon source, Nitschke et al. [67] found 10 rhamnolipid homologues produced by LBI. Lovaglio et al. [51] reported the production of nine rhamnolipid types by *Pseudomonas aeruginosa* LBI 2A1 using sunflower oil, castor oil and corn oil sludge. The variation in the proportion of homologues produced depends on the strain, culture medium, cultivation conditions and culture time [68]. This variation in the composition influences the physicochemical properties of rhamnolipids [57,69]. With the use of guava seed oil as the carbon source, a homologue with a molar mass of 555.6 (RhaC$_{10}$C$_{14:2}$) appeared, which has not previously appeared with the other carbon sources used by *P. aeruginosa* LBI 2A1, thus agreeing with Mata-Sandoval et al. [68]. Using glycerol as a carbon source for *P. aeruginosa* LBI 2A1, Salazar-Bryam et al. [40] also found that the congeners were maintained in relation to other carbon sources used.

More than 58 homologous species have been described in the literature, and these can be classified into the groups of mono-rhamno-mono-lipids, mono-rhamno-di-lipids, di-rhamno-mono-lipids and di-rhamno-di-lipids. -lipids. The variation of structures within these groups is generally related to the amount of carbons (lengths ranging from C8–C16) and the presence and amount of unsaturation in the aliphatic chain of β-hydroxy acids [33]. In cases of isomeric species of RLs that contain two fatty acid moieties with different lengths, there is a greater predominance of the isomer that contains the shortest chain linked to the hydrophilic moiety. This occurrence is at least twice as abundant and is even greater when one of the fatty acid portions is unsaturation, where the preference is that the smallest saturated chain is adjacent to the rhamnose [65,70]. This trend was observed in homologous species of mono-rhamnolipids, in ions 475.6, 529.2, 530.9 and 555.6. These molecules present aliphatic chains of different sizes and the fragmentation spectra indicated the smaller chains linked to rhamnose.

## 4. Conclusions

*Pseudomonas aeruginosa* LBI 2A1 was able to metabolize guava seed oil and produce high concentration of rhamnolipids, with a predominance of mono-RL probably because of the long chain fatty acid composition. The experimental design showed that nitrogen concentration was the most significant variable to influence the biosurfactant production and the optimal concentrations of carbon and nitrogen source was 100 and 10 g/L, respectively. RL characterization showed 9 homologues species with one compound (*m/z* 555) not yet described in the literature. In conclusion, this research showed that the use of guava seed oil combined with experimental design are great strategies for rhamnolipids production with a rare homologue. However, the continuity of the experiments proves to be important for further studies, such as studying production in reactors for scale-up, the influence of the guava seed oil composition in the homologous species, such as their purification and evaluation in cosmetic applications.

**Supplementary Materials:** The following supporting information can be downloaded at: https://www.mdpi.com/article/10.3390/fermentation8080379/s1, Figure S1: Fragmentation of rhamnolipid ions by ESI (-) MS/MS from RL samples by guava oil.

**Author Contributions:** Conceptualization, J.C.; methodology, I.Y., A.M.S.-B., A.U.d.F. and L.P.L.; investigation, I.Y. and A.U.d.F.; writing—original draft preparation, I.Y.; writing—review and editing J.C.; supervision, J.C.; data curation, R.B.L. and J.C.; funding acquisition, J.C.; project administration, J.C. All authors have read and agreed to the published version of the manuscript.

**Funding:** This research was funded by Brazilian fostering agencies National Council for Scientific and Technological Development (CNPq), grant 302367/2019-5, Coordination for the Improvement of Higher Education Personnel (CAPES), grant 88887.480773/2020-00 and São Paulo Research Foundation (FAPESP), grants 2020/06189-1 and 2017/22401-8.

**Institutional Review Board Statement:** Not applicable.

**Data Availability Statement:** Not applicable.

**Acknowledgments:** The authors would like to express sincere thanks to Renan Pirolla for his support in the LC-MS analysis.

**Conflicts of Interest:** The authors declare no conflict of interest. The funders had no role in the design of the study; in the collection, analyses, or interpretation of data; in the writing of the manuscript; or in the decision to publish the results.

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
