# Peer review of "Guava Seed Oil: Potential Waste for the Rhamnolipids Production"

_fermentation, doi:10.3390/fermentation8080379_

Round 1
Reviewer 1 Report
Report
The manuscript entitled “Guava seed oil: potential waste for the rhamnolipids production” by Yoshimura et al described a study on growth of Pseudomonas aeruginosa LBI 2A1 to produce rhamnolipids using guava seed oil as carbon source. Rhamnolipid is one of the most studied biosurfactants that are commonly produced by P. aeruginosa cultivated in growth medium supplemented with carbon sources such as glucose, glycerol, and oil / lipids. Indeed, using byproducts such as frying oil and residue glycerol without the need of further processing can greatly reduce the cost of carbon source in rhamnolipid production. In case of guava seed oil, it appears that the authors need to perform oil extraction from seeds before use. This is not a simple process: one can use the same process to extract oils from common oil crops. If this is meant for disposal of guava peels and seeds as waste, why not use it to produce guava seed oil? It is known to have many health benefits.
Major concern
The authors need to justify the use of guava seed oil as carbon source in rhamnolipid production for reducing production cost. Guava seed oil in market is much more expensive than many other common cooking oils.
Minor concern
Line 92, remove redundant word “syrups”
Line 180, use proper degree sign “°” as in line 173
Line 189, use subscript “3” in formula BF3
Line 195, use subscripts “2” and “4” in formula Na2SO4
Line 199, change “/min” to “min-1”, in which use superscript “-1”, recommend to change the format throughout the manuscript.
Author Response
Replies to Reviewers
First, we want to thank the reviewers for their suggestions, which help to improve the article.
REVISOR 1:
Regarding the question about the use of guava seed oil, we put below the explanation of why it is used:
The use of guava seed is part of a larger project called fructorefinery. The national fruitculture is one of the most important sectors of Brazilian economy, with significant contributions to the production of fruits, industrialized juices and sweets, exportations of industrialized food, and large-scale employment. Despite its great importance for the national economy, the agroindustry generates huge amounts of residues, which contribute to increasing problems related to environmental issues. For this reason, our current project has the major aims: to propose rational uses for residues derived from the fruitculture industry and to generate high-value products. This project intends to transform agroindustrial residues from fruits such as orange, banana, mango, pineapple and guava, by using processes such as microbial fermentation, enzymatic hydrolysis, auto-hydrolysis, and bioelectrochemistry, and by applying new methods for characterization of raw materials and their respective derived high-value compounds. Therefore, in the case of guava, after extracting the oil, there is still the residual of the seed that will be hydrolyzed to obtain other metabolites. So, in addition to giving a destination to the waste that would be destined for the landfill, we value it through obtaining molecules. The use of the oil is also intended to produce congeners of rhamnolipids that have not yet been identified. It is not just a matter of cost, but of achieving what we call sustainability, contributing to showing that there is one more use for the waste that would be discarded. In addition to the above justification, it should also be added that the contribution towards the reduction of the cost of a biotechnological product is not only given through the cost of the carbon source, but also in relation to the production of rhamnolipids reached and this oil generate a high concentration of this metabolite.
Regarding the questions with minor concern:
We redid the parts according to the suggestions, with the exception of the last item, where it was suggested to change the format of the unit of measure: Line 199, change “/min” to “min-1”, in which use superscript “-1”, recommend to change the format throughout the manuscript. Only in this case, we saw no need for change.
Reviewer 2 Report
The manuscript has attention to the valorisation of the guava seeds.
The authors are kindly requested to address the following recommendations:
- improve the introduction by avoiding the repetition of the word biosurfactant(s) and reducing the extensive use of definitions/basic information;
- write the species names in italics in the entire manuscript;
- mention if Trials 9-12 (line 253) correspond to E9-E12 from the table;
- mention the abbreviations names where they are firstly appearing (for example, RL);
- consider discussing your results with similar results already published in the literature
- under the Conclusion section, highlights the main achievements and insert future perspectives.
Author Response
Replies to Reviewers
First, we want to thank the reviewers for their suggestions, which help to improve the article.
REVISOR 2
We redid the parts according to the suggestions.
Regarding the section indicating improvement in English, we are sending the certificate that the article has been reviewed by a native English speaker.

Round 2
Reviewer 1 Report
In response to reviewers’ commend, the authors stated that utilization of guava seed oil is part of the national project known as fructorefinery in Brazil. This explanation should be included in the Instruction section of the manuscript, simply because that no novel methodologies are involved in the production and analysis of rhamnolipids in the manuscript, except for usage of guava seed oil as carbon source.
Reviewer 2 Report
The authors have satisfactorily addressed the recommendations of the reviewer.